# Development of Washable Silver Printed Textile Electrodes for Long-Term ECG Monitoring

**DOI:** 10.3390/s20216233

**Published:** 2020-10-31

**Authors:** Abreha Bayrau Nigusse, Benny Malengier, Desalegn Alemu Mengistie, Granch Berhe Tseghai, Lieva Van Langenhove

**Affiliations:** 1Department of Materials, Textiles and Chemical Engineering, Ghent University, 9000 Gent, Belgium; Benny.Malengier@UGent.be (B.M.); GranchBerhe.Tseghai@UGent.be (G.B.T.); Lieva.VanLangenhove@UGent.be (L.V.L.); 2Ethiopian Institute of Textile and Fashion Technology, Bahir Dar University, Bahir Dar 6000, Ethiopia; dmengist@calpoly.edu; 3Materials Engineering Department, California Polytechnic State University, San Luis Obispo, CA 93407, USA; 4Jimma Insitute of Technology, Jimma University, Jimma P.O. Box 378, Ethiopia

**Keywords:** ECG, conductive textiles, textile electrodes, washable electrodes

## Abstract

Long-term electrocardiography (ECG) monitoring is very essential for the early detection and treatment of cardiovascular disorders. However, commercially used silver/silver chloride (Ag/AgCl) electrodes have drawbacks, and these become more obvious during long-term signal monitoring, making them inconvenient for this use. In this study, we developed silver printed textile electrodes from knitted cotton and polyester fabric for ECG monitoring. The surface resistance of printed electrodes was 1.64 Ω/sq for cotton and 1.78 Ω/sq for polyester electrodes. The ECG detection performance of the electrodes was studied by placing three electrodes around the wrist where the electrodes were embedded on an elastic strap with Velcro. The ECG signals collected using textile electrodes had a comparable waveform to those acquired using standard Ag/AgCl electrodes with a signal to noise ratio (SNR) of 33.10, 30.17, and 33.52 dB for signals collected from cotton, polyester, and Ag/AgCl electrodes, respectively. The signal quality increased as the tightness of the elastic strap increased. Signals acquired at 15 mmHg pressure level with the textile electrodes provided a similar quality to those acquired using standard electrodes. Interestingly, the textile electrodes gave acceptable signal quality even after ten washing cycles.

## 1. Introduction

The demand for wearable electronic systems is increasing due to the growing interest of users in taking active control of their health as part of a preventive lifestyle and other applications [1,2]. Wearable products contain various types of sensors integrated in watches, wrist bands, belts, garments, and textiles or directly applied on the skin. Sensors for health-related applications are capable of monitoring physiological parameters like electrocardiography (ECG) [3], electromyography [4], electroencephalography [5], temperature [6], and respiration rate [7]. Among these, ECG monitoring is the most common application area of textile-based sensors. ECG is the process of recording the electrical activity of the heart and is one of the most widely used health monitoring methods to diagnose and examine the development of cardiovascular diseases (CVD), which are the leading cause of death worldwide [8,9,10]. Continuous long-term ECG monitoring is very important for early detection and treatment of CVD before they evolve into serious complications [11]. During ECG monitoring, the change in the electrical activity of the heart over time is detected and converted into a signal, which then divulges information about the health condition [12,13]. Any ECG device contains three essential components: electrodes, interconnection wires, and a processing unit. Of these, the electrodes are the main components that affect the quality of the acquired signal. Disposable silver/silver chloride (Ag/AgCl) gelled electrodes are most commonly used for ECG signal measurements. However, the adhesive gel used in Ag/AgCl electrodes dehydrates gradually during usage, the signal quality degrades over time and skin irritation occurs. Such electrodes are not suitable for continuous monitoring because their drawbacks become more obvious during long-term signal monitoring [14,15]. Dry electrodes made from textiles are a potential choice for long-term signal monitoring because of their air and water permeability, which makes users feel more comfortable wearing them [14].

Electrically conductive textiles can be directly used as ECG electrodes. Different types of textile-based electrodes have been developed by knitting [16], weaving [17], and embroidering [18] conductive yarns to conventional fabric or by screen printing [19], plating [20], or dip coating [21] conductive material over the fabric surface. The resulting conductive textile can work based upon the non-contact or contact principle. In non-contact electrodes, there is no direct contact with the skin; the electrodes are separated from the skin by a dielectric material with a high dielectric constant [10,22,23]. The electrodes can be integrated into a stretcher, hospital beds, and wheelchairs [24], mattresses, and pillows [22], chest belts [25], or t-shirts [23]. Such electrodes are ideal for wearable ECG monitoring due to the absence of direct contact with skin, but there is a high difficulty in controlling the capacity change during the movement of the user, which limits their application [26]. Contact-based electrodes work through direct contact between the skin and the electrode, which is essential to create a conductive path where the ECG signal can be recorded by placing conductive electrodes at different locations on the body [27,28]. Such electrodes are required to have conformal contact with the skin of the wearer for continuous and constant signal detection and to minimize unwanted signals, called artifacts [29].

Polyester fabric coated with nickel, copper, and gold used as a wrist band for ECG monitoring has been presented by Das et al. [20]. They reported that the electrodes acquired an adequate ECG signal from the wrist when the subject is at rest, but the ECG was inadequate when recorded during dynamic conditions. Metal-coated conductive textile materials are attractive for biopotential monitoring due to their high conductivity; however, their development techniques are not easy.

Even though textile-based electrodes are preferable for long time ECG monitoring, they are sensitive to motion artifacts due to the movement of electrodes on the skin surface while the person is in dynamic condition [30,31]. To overcome such problems and to fix the electrodes in place irrespective of the user’s movement, the electrodes are embedded in stretchable materials that exert some amount of holding pressure. The level of the applied pressure on the body affects the signal quality and helps to have a clear signal, but too high a pressure causes discomfort and can even affect the blood circulation [31,32]. Tong et al. studied the sensitivity of the textile-based electrodes to skin-textile contact at various holding pressure. ECG recording was performed by varying the applied pressure from 12 to 60 mmHg to vary the skin–textile contact. Results showed that better signal quality is obtained when the pressure is ≥ 36 mmHg, however, high tightness brings discomfort especially if the pressure is over 30 mmHg [33]. Ideally, a stretch fabric does not exert more than 10 mmHg, and hence research is ongoing for good electrodes operating around this pressure level.

Several issues with regards to textile-based ECG remain unsolved, such as the development techniques that are compatible with the conventional textile production process, washing durability, minimizing unwanted signals, and how to integrate them into wearable systems. Ankhili et al. presented a wearable ECG monitoring bra that employed poly(3,4-ethylenedioxythiophene) polystyrene sulfonate (PEDOT:PSS) printed on cotton, polyamide, and polyester electrodes and evaluated the performance of signal quality after 50 washing cycles. They reported that the cotton and polyamide electrodes provided good signal quality, while the signal collected by PEDOT:PSS printed polyester electrodes showed lower signal quality, where major peaks were not identifiable [9].

In this work, we developed textile electrodes for ECG monitoring by screen-printing silver ink on cotton and polyester knitted fabrics. Knitted fabric was preferred because of its stretchability and ability to fit the body. The screen-printing technique was selected because it is easy to adopt, relatively cheap process and it is compatible with the conventional roll-to-roll processing of textiles [34,35]. For materials that come into direct contact with human skin, like bio-potential electrodes, the materials are required to be biocompatible. In this case, silver ink is chosen, which is corrosion-resistant, bactericidal, has anti-allergic properties, and has less potential to trigger skin irritation with good electrical stability when exposed to sweat [31]. The aforementioned properties of silver make silver printed electrodes suitable for long-term ECG monitoring. ECG signals were collected from the wrist while sitting and in walking positions at different skin-electrode contact pressures before and after multiple washing. The acquired ECG signals were comparable with signals from standard Ag/AgCl electrodes. As silver is well suited for bioelectric measurements and has good characteristics [36], we focus in this work on the quality of the actual ECG measurements. In addition to the electrode’s electrical conductivity, the skin-electrode impedance measurement is very important to evaluate whether the electrodes can be used for recording bioelectric signals; too high skin-electrode impedance means the electrodes cannot be used for ECG recording. However, as this test requires separate measuring devices, we plan to do this in our follow up works.

## 2. Materials and Methods

### 2.1. Materials

Two types of knitted fabrics were selected, 160 grams per square meter (GSM) cotton, and 140 GSM polyester fabric. Silver ink (Metalon HPS-FG32) having a solid content of 75% and particle size 1.5 micron obtained from Novacentrix, was used to develop the electrodes. This ink is compatible with textile fabric and the screen-printing technique, and its high silver content is important to produce a fabric with high electrical conductivity.

### 2.2. Textile Electrode Development

Home-developed flat screen-printing (with mesh count 90) (Figure 1a) was used to print the electrodes onto the two knitted fabrics. For all samples, the squeegee passed over the screen five times to ensure penetration of the ink through the fabric substrate. Three samples with an electrode size of 6 cm^2^ (3 × 2 cm^2^) were printed from each knitted textile substrate. Curing was carried out in an oven dryer at 120 °C for 30 min, as recommended by the manufacturer. Samples were conditioned in a standard atmosphere room for 24 h (RH = 65 ± 2% and T = 20 ± 2 °C) before measuring their electrical conductivity. A metallic snap was attached to the textile electrode to connect to the wires of an ECG recording module, as shown in Figure 1b.

### 2.3. Electrical and Microscopy Characterization

The surface resistance of printed electrodes was measured via the two-point probe method using a Fluke 87 multimeter. At least three samples were measured (five readings for each sample at different measurement points) to compute the mean and standard deviation (SD). The surface image of the textile fabric before and after printing was investigated by scanning electron microscope with an acceleration voltage of 20 kV (SEM- FEI Quanta 200 FFE). The uncoated fabrics were sputtered with gold to avoid the charging effect.

### 2.4. ECG Measurements and Signal Evaluation Method

The electrodes were embedded on an elastic strap with hook and loop fasteners (Velcro), which have adjustable length to securely attach the electrodes on various wrist sizes at different holding pressures. ECG signal was collected from the wrist with lead I configuration, where the measuring electrodes were placed on the right and left wrist, and the reference electrode was placed on the backside of the left forearm close to the elbow (as shown in Figure 1c). The edge to edge distance between the measuring electrode and the reference electrode on the left forearm was 11 cm. It is known that the outer layer of the skin (strateum corneum) has resistive nature, and is removed during skin preparation before caring out ECG measurement. This skin preparation needs experts and is time-consuming, which makes it challenging for individuals who need to measure their ECG in their homes. Dry electrodes are expected to use without any prior skin preparation. No skin preparation was done and an ECG was taken 5 min after attaching the electrode to the subject’s skin. ECG measurement was conducted at a sitting position with two arms on the table, as well as while walking on a smooth surface. One volunteer with no heart problem history participated in the ECG measurement. The subject was informed to avoid unnecessary movement while measuring ECG, to minimize the effect of body movement on the signal quality. PC-80B easy ECG monitor (Shanghai Lishen Scientific Equipment, Shanghai, China), a portable ECG device with an ECG bandwidth of 1–40 Hz and heart rate (HR) measurement accuracy of ±2 bpm was used to collect ECG. Data were recorded for 3 min in every test and the stored data uploaded to a computer for analysis using the ECG viewer manager provided by the manufacturer.

To investigate the effect of strap tightness on stability and signal quality, ECG was measured from 6 to 15 mmHg contact pressure with a 3 mmHg interval. This was achieved by adjusting the Velcro, where the pressure measurement was carried out at the particular location where the electrodes were attached. The holding pressure was measured using a Microlab PicoPress instrument M-1200. We also collected ECG using standard electrodes (Ag/AgCl) for comparison, with asynchronous ECG recording method, i.e., both types of electrodes fixed at the same place and signal recorded at different times. Though there are time-based differences in this approach, it is better to allow comparison, as holding pressure and environment variables should be comparable [37]. To prevent time-based variation, some researchers use synchronous (placing the two types of electrodes at the nearest possible location) and measure the ECG with two devices, but this causes position-induced variation which also causes differences in the comparison as well as possible ECG device differences [31,37]. For this reason, we selected the asynchronous approach. Signals were first measured using Ag/AgCl electrode and cleaned after testing to remove gel residuals on the skin, then textile electrodes are placed exactly at the same place to carry out the next measurement. The subject takes a 5-min break after every measurement before going to the next test.

Signals collected by each type of electrode were evaluated based on visibility and amplitude of P, R, and T peaks and signal intervals like PR, QT, and QRS intervals. The amplitude of each peak and noise was determined from the ECG viewer manager software (V5.2.0.1). Signal to Noise Ratio of the measured voltage (SNR voltage) was calculated as the ratio of the QRS amplitude (QRSamp) to the amplitude of the noise (Noiseamp), as shown in Equation (1) by modifying Tsukada’s method [38], which is normally reported as a signal to noise ratio (SNRdB) derived from SNR voltage based on Equation (2). The amplitude of the noise was extracted from a section of each cardiac cycle by determining the magnitude of the peak and lowest valley other than PQRST components in that section. Signal to Noise Ratio of the measured voltage (SNR voltage) was then calculated via Formula (1) and (2).
(1)$$\mathrm{SNR}\;(\mathrm{voltage})\;=\;\left|{\mathrm{QRS}}_{\mathrm{amp}}/{\mathrm{Noise}}_{amp}\right|$$
SNR_dB_ = 20 log_10_ (SNR (voltage))(2)

To assess the ECG performance of the electrodes between individuals, tests were performed with six subjects (three males, three females, age 28 ± 8) with no heart problem history. Collected signals were compared based on the time between subsequent R peak and (R-R interval) and average HR. All ECG measurements were undertaken based on a protocol approved by the Institutional Review Board of EiTEX, Bahir-Dar University.

### 2.5. Washing, Environmental, and Stretching Durability Tests

To investigate the washability performance of the electrodes, the samples were washed up to 10 washing cycles in a domestic washing machine. The printed electrodes were washed with 4 g/L non-ionic detergent at 40 °C for 30 min according to the standard method (ISO 105-C06-A1S, 2010). The samples were next dried at room temperature. This process was then repeated for each washing cycle. Three samples were washed from each type, which is silver printed cotton and silver printed polyester.

The effect of stretching on the electrical conductivity of the textile electrodes was also studied by stretching the textile electrodes on a light-duty drill vise modified for this purpose. The samples were stretched up to 30% of its original length with 5% intervals and the surface resistance was measured, three samples were tested for each electrode. The relative resistance ratio was computed using (R_i_/R_0_), where R_0_ is the surface resistance of the sample at no stretching and R_i_ is the surface resistance at the ith stretch percentage.

Additionally, the environmental stability of the electrodes was studied by keeping the printed electrodes in the ambient environment for six months. The ECG detection performance was evaluated for the electrodes stored for six months in the same manner.

## 3. Results and Discussion

### 3.1. Electrical and Morphological Properties

The electrical conductivity of the electrodes is one of the factors that affect the ability of the electrodes to collect a biosignal. As the resistance of the electrodes becomes lower, the ability to collect signals and its stability becomes better [26,39]. The surface resistance was 1.64 ± 0.09 Ω/sq and 1.78 ± 0.14 Ω/sq for the silver printed cotton and the polyester electrode, respectively. An independent t-test was done, and the results showed that there is a statistically significant difference in surface resistance between the two types of electrodes (*p*-value = 0.003, alpha = 0.01). The difference in surface resistance from one measurement point to another is very small, as is evident through the small standard deviation, which shows uniform deposition of the silver ink on the fabric surface through the screen-printing method. Our silver printed textile electrodes have higher conductivity than PEDOT:PSS coated textile electrodes; for example Trindade et al. reported PEDOT:PSS coated plain weave fabrics of polyester electrode with a surface resistivity of 10 Ω/sq for ECG monitoring [40].

Figure 2 shows the SEM images of the cotton and polyester fabric before and after silver ink printing (the top images show low magnification and the bottom higher magnification). It is clearly seen that cotton is densely and homogeneously coated with silver flakes owing to its relatively rough surface due to the presence of convolutions and the hydrophilic property. This dense and homogenous coating is the main reason for the higher conductivity of cotton electrodes. The silver flakes were seen to have a size ranging from less than 1 to 2 μm, which is in agreement with the supplier information (inset in Figure 2c).

### 3.2. ECG Signal Acquired Using Textile Electrodes

The ECG signal collected using Ag/AgCl electrodes is presented in Figure 3c, while Figure 3a,b show ECG signals collected by silver printed cotton and polyester electrodes, respectively, at the same holding pressure of 15 mmHg. The results clearly reveal that the three major peaks are visible in all ECG signals collected using cotton and polyester electrodes and the quality of the signal is comparable to the signal collected by gelled electrodes. All peaks, i.e., P, QRS, and T waves are visible. Signals without any missing R-peaks and no falsely detected R-peaks due to interferences are considered proper ECG signals [41]. The signals collected by silver printed textile electrodes showed clear R-peaks without any missing and false peaks, which indicates that the waveforms are acceptable.

The interval between two consecutive R-peaks also called the R-R interval in the ECG signal is used to estimate heart rate [42]. The average HR of the signal taken from the gelled electrode was 76 bpm, while the asynchronous result from the textile electrodes was 75 and 73 bpm for cotton and polyester electrodes, respectively, which is easily explained by the asynchronous measurement. The standard value of HR for men is in the range of 60–100 bpm [39] and all the results were in this range. All the signals have approximately the same R-R interval without any irregular rhythm type.

The small-amplitude waveforms, i.e., P and T waves were almost equal for all electrodes, and were 0.09 and 0.35 mV, respectively, as shown in Table 1. The mean and standard deviation of each parameter was calculated from the values of twenty-five consecutive cardiac cycles. The signal from cotton has a higher R-peak amplitude than the polyester electrode, and the values were 1.28 and 1.26 mV, respectively, while for the standard Ag/Ag Cl electrode the R-peak was 1.29 mV. The QRS interval was almost equal for both the textile and Ag/AgCl electrodes at 93 ms. The standard value of QRS is 60–100 ms [43]. The time for the PR interval was in the standard range of 120–200 ms [44], specifically 158 ms for cotton and gelled electrodes and 154 ms for polyester. The QT interval range was 346 and 336 ms for cotton and polyester electrodes, respectively, while the expected value is < 400 ms [43]. Signals acquired by Ag/AgCl had 348 ms as the QT interval.

Signal noises are unwanted signals caused by external factors; good signals give a high SNR. The SNR was 33.1 dB for signals acquired using the cotton electrode, while it was 33.52 dB and 30.17 dB for signals acquired using gelled electrode and silver printed polyester electrodes, respectively. There is no clear difference in waveform between the signals collected from cotton and standard Ag/AgCl electrodes (Table 1). Kannaian et al. also reported embroidered ECG electrodes with 31.6 dB SNR [18].

We have also assessed the ECG performance of the electrodes between individuals by performing tests with six subjects. Table 2 shows the average HR and R-R interval values of the three electrodes for each subject that participated in the study. The results revealed that the difference between HR and R-R interval values acquired using the three electrodes is small among all individuals.

### 3.3. Effect of Holding Pressure Level on ECG Signal Quality

The signal stability and quality of dry electrodes are significantly affected by skin-electrode contact pressure [33]. Figure 4 shows ECG signals acquired using the silver printed cotton electrodes at different holding pressure levels. Signals acquired using the textile electrodes at 6 and 9 mmHg pressure showed fluctuations in R-peak amplitude and the P-waves are not clear enough, which is due to poor skin-electrode contact. ECG signals collected at a contact pressure of 12 mmHg and 15 mmHg provided clear cardiac waves and even P waves were distinguishable in all signals with these holding pressures. Silver printed polyester textile electrodes also showed the same trend (results not shown here). As pointed out earlier, the ECG signals acquired using silver printed cotton electrodes at 15 mmHg pressure level have a similar quality to the signals collected using standard Ag/AgCl electrodes. The amplitude of the R-peak of the signals collected at a pressure level of 12 and 15 mmHg has no significant difference with signals from Ag/AgCl (*p*-value = 0.423 and 0.963 at alpha = 0.01). This could be due to the decrease in contact impedance as contact pressure increases, which enables an easy flow of current at the skin-electrode interface and results in better signal quality [31]. The gel used in standard disposable electrodes can maintain good skin-electrode contact without any external holding pressure.

Figure 5 shows the comparison of the R-peak amplitude of signals for silver printed cotton textile electrodes at different holding pressure levels and the standard Ag/AgCl electrode. The maximum R-peak amplitude for the silver printed cotton electrode was 1.24, 1.32, 1.40, and 1.38 mV, whereas the mean was 1.14, 1.24, 1.26, and 1.28 mV at 6, 9, 12, and 15 mmHg holding pressure levels, respectively. For the standard Ag/AgCl electrode, the maximum R-peak amplitude was 1.40 mV and the mean was 1.29 mV. The results showed that, as the holding pressure increases, the R-peak amplitude of the signals increases, that is, a better signal is captured.

### 3.4. Effect of Body Motion on the ECG Quality

It is expected that due to difficulty in keeping the electrodes attached at a constant place on the user’s body, the signal quality drops while the subject is walking. This is the key problem for dry textile electrodes and even gelled Ag/AgCl electrodes as well. Tong et al. also reported that ECG signals collected from textile electrodes when the subject is walking had a lower quality, where the small amplitude peaks were not identifiable [33]. Nevertheless, in our work, good signals were still acquired by the textile electrodes at 15 mmHg when the body was in motion (Figure 6). This is due to the pressure exerted by the elastic strap, which increased the contact between the textile electrode and the body. The cotton electrodes show relatively better signal quality compared to polyester electrodes during this dynamic condition. The SNR of signals collected at walking using silver printed cotton electrodes was 20.8 dB, while it was 18.9 and 25.2 dB for signals obtained using silver printed polyester and standard Ag/AgCl electrodes, respectively. Improving the signal quality of the textile electrode during dynamic conditions is of great importance and work is in progress. The first aim here is to verify that enough good measurements are possible per minute during movement to still obtain a qualitative view of the heart condition.

### 3.5. Washing and Stretching Durability

Figure 7a presents the surface resistance of the electrodes after multiple washing cycles. After one washing cycle, the surface resistance increased to 3.14 ± 0.16 Ω/sq (1.9 times the original value) for cotton and 3.77 ± 0.25 Ω/sq (2.12 times the original value) for polyester. We assume that this is because loosely attached silver flakes were washed out from the fabric surface. Even though there is a small increase in surface resistance after five washing cycles in the cotton electrodes, the electrodes still had acceptable electrical conductivity to be used for ECG signal acquisition. However, in polyester, the surface resistance increases fast reaching nine times the original value after ten washing cycles, while this was only 3.72 times for cotton. The cotton electrode is hence much more stable under washing. The mechanism of attachment of silver flakes to the fabrics is not clear, as the additive used in the silver ink is not revealed. The better wash durability of silver printed cotton electrodes could partly be due to the denser packing of the flakes and perhaps better attachment between the convoluted rough surface cellulose of cotton and the silver ink.

The stretch test results also show that there is only a slight increase in resistance with stretching up to 30% (Figure 7b). The resistance of silver printed cotton increased by only 1.06 and 1.23 times for silver printed cotton and polyester electrodes, respectively at a 30% stretch percentage. This better performance during stretching of silver printed cotton electrodes than polyester electrodes is in agreement with the wash durability of the electrodes.

### 3.6. ECG Signal Acquired after Electrode Washing

To investigate the suitability of the electrodes for a long-term wearable ECG monitoring system, the ECG signal acquired by the cotton and polyester electrodes after washing was evaluated. Even though there was a small absolute increase in surface resistance after five washing cycles, the electrodes still had acceptable electrical conductivity to be used for ECG signal acquisition, and even after ten washing cycles for cotton, as shown in Figure 8.

Signals acquired after one washing cycle (Figure 8a) do not show any signal deterioration, i.e., all the major cardiac peaks are clearly recognizable. Similarly, good signals could be obtained after five (Figure 8b) wash cycles, but after ten washing cycles (Figure 8c), the amplitude of the R-peaks becomes significantly lower than the non-washed samples (*p*-value < 0.001, alpha = 0.01), with a magnitude of 0.05, 0.27 and 0.86 mV for P, T and R peaks, respectively. This is because of the increase in surface resistance which leads to an increase in skin-electrode impedance. In the unwashed case, the R-peak was 1.28 mV (see Table 1).

To study the stability of the electrodes for long-term use, the ECG signal was collected by the silver printed cotton and polyester electrodes after storing the electrodes for six months at ambient atmosphere. The electrodes did not show a difference in ECG detection performance even after six months. As shown in Figure 9, all the major picks are identifiable. The ECG signals collected from the embroidered textile electrodes have an identifiable P, QRS, and T waveforms with amplitude 0.09, 0.30, and 1.27 mV for P, R, and T waves, respectively. There is no significant difference in the R-peaks amplitude (*p*-value = 0.89 and alpha = 0.01). The average HR was 76 and 71 bpm for cotton and polyester, respectively, which is almost the same as the original experiment.

To assess the skin reaction during ECG testing, volunteers were asked about and did not report any discomfort during the test itself. A visual inspection of the skin was done to ensure there was no visible damage or skin coloration. None of the tests showed visual changes to the skin, which is to be expected, as silver has less toxicity to skin [45].

## 4. Conclusions

Screen printing of silver ink on knitted cotton and polyester fabrics was used to develop textile electrodes for continuous ECG measurement. The surface resistance of the silver printed cotton and polyester was 1.64 and 1.78 Ω/sq, respectively. ECG signals collected using the dry textile electrodes showed clear R-peaks without any missing and false peaks, which are comparable with the standard Ag/AgCl gel electrodes. The amplitudes of P and T waves were 0.09 and 0.035 mV, respectively, and were equal for all electrodes. The R-peak amplitudes were 1.28 and 1.26 mV with SNR of 33.10 and 30.17 dB for cotton and polyester electrodes, respectively, and 1.29 mV and 33.52 dB SNR for the standard Ag/AgCl electrode, showing that the cotton electrode is comparable to the standard electrode, while the polyester electrode showed slightly inferior values due to its slightly higher surface resistance. The results revealed that the ECG signal quality acquired using dry textile electrodes increased as the holding pressure increased. Signals collected at 15 mmHg pressure level have a similar quality to signals collected using standard Ag/AgCl gel electrodes. The silver printed cotton electrodes gave acceptable waveforms after 10 washing cycles even though the surface resistance increased slightly. The textile electrodes were also environmentally stable showing no difference in ECG detection performance when a measurement was taking after storing the electrodes for six months in an ambient atmosphere. The signal quality was low when the measurement was done in a moving body. The signal quality in motion can be improved by increasing the holding pressure of the textile electrodes exerted by using elastic straps. The developed electrodes would be functional for portable ECG devices to record ECG and determine heart rates in non-dynamic conditions like people sitting or sleeping. Follow-up research should focus on improving the signal quality of the textile electrode under movement to allow the development of a true textile wearable system for long-term ECG monitoring.

## Figures and Tables

**Figure 1 sensors-20-06233-f001:**
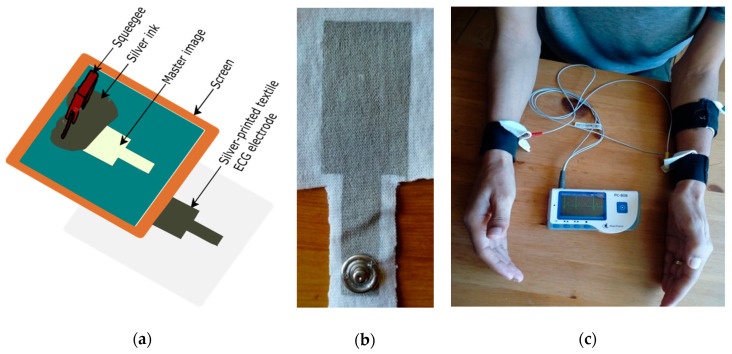
(**a**) Block diagram for screen printing process; (**b**) Front view of the developed electrode; (**c**) Component placement setup for ECG measurement.

**Figure 2 sensors-20-06233-f002:**
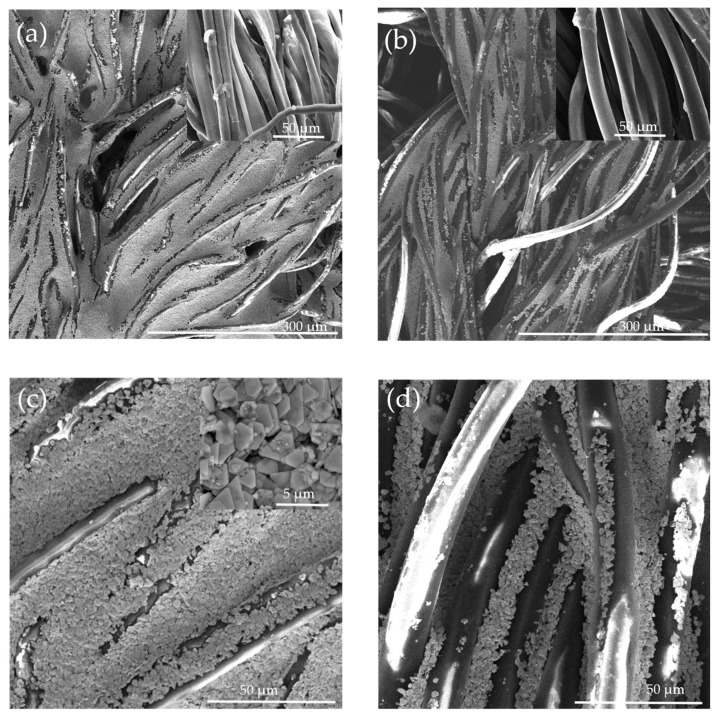
SEM image of (**a**,**c**) silver printed cotton fabric and (**b**,**d**) silver printed polyester fabric at different resolutions. Insets in the upper right part of (**a**,**b**) show the non-printed plain knitted fabrics, while the inset of (**c**) shows a high magnification of the silver particles on cotton.

**Figure 3 sensors-20-06233-f003:**
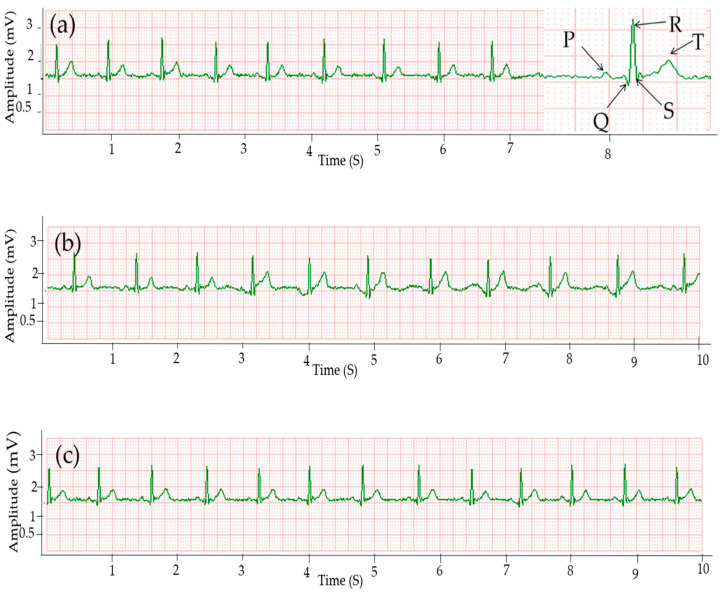
ECG signals collected using: (**a**) silver printed cotton; (**b**) silver printed polyester fabric; and (**c**) Ag/AgCl gel electrode.

**Figure 4 sensors-20-06233-f004:**
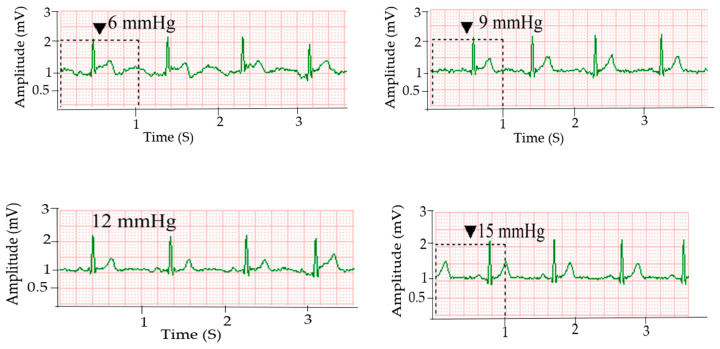
ECG signals collected using silver printed cotton electrode at different pressure levels.

**Figure 5 sensors-20-06233-f005:**
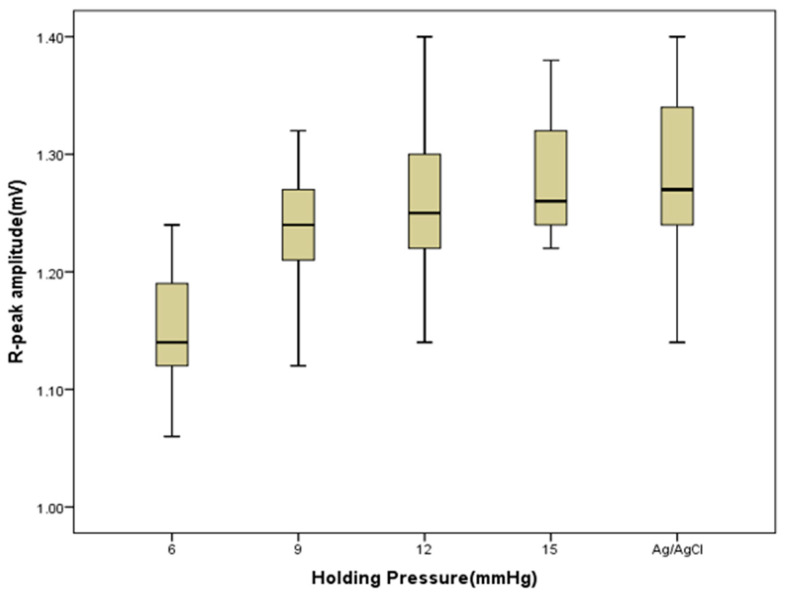
Boxplot of the R-peak amplitude of signals obtained from silver printed cotton electrodes with different holding pressure, and Ag/AgCl electrode as a control.

**Figure 6 sensors-20-06233-f006:**
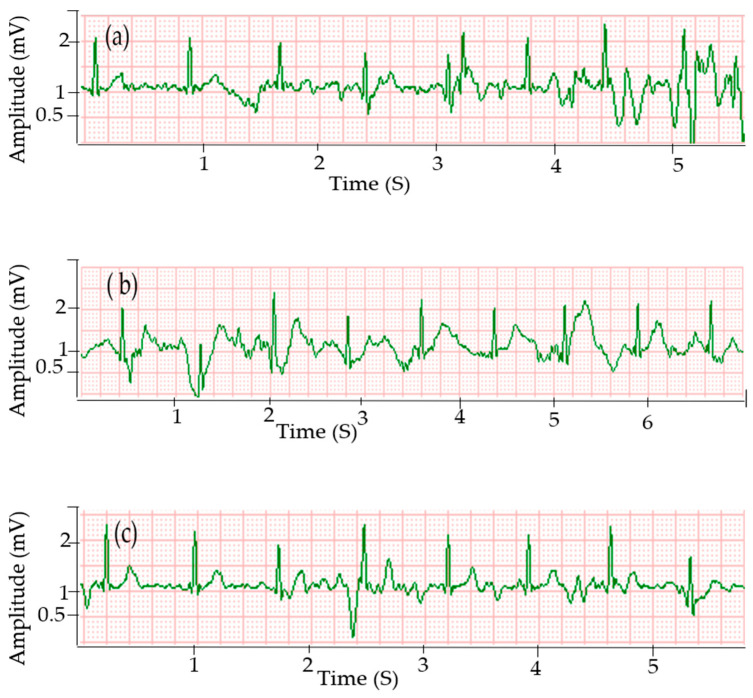
ECG signals acquired when the body is in motion using different electrodes: (**a**) silver printed cotton at 15 mmHg; (**b**) silver printed polyester at 15 mmHg and (**c**) Ag/AgCl electrodes.

**Figure 7 sensors-20-06233-f007:**
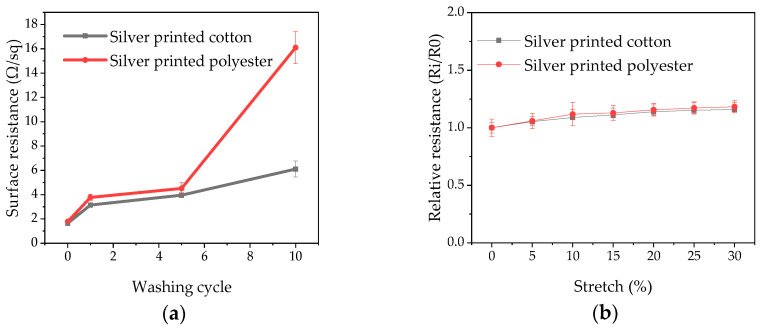
(**a**) Change of surface resistance of silver printed textile electrodes with multiple washing cycles; (**b**) Effect of stretching on-resistance of printed fabric.

**Figure 8 sensors-20-06233-f008:**
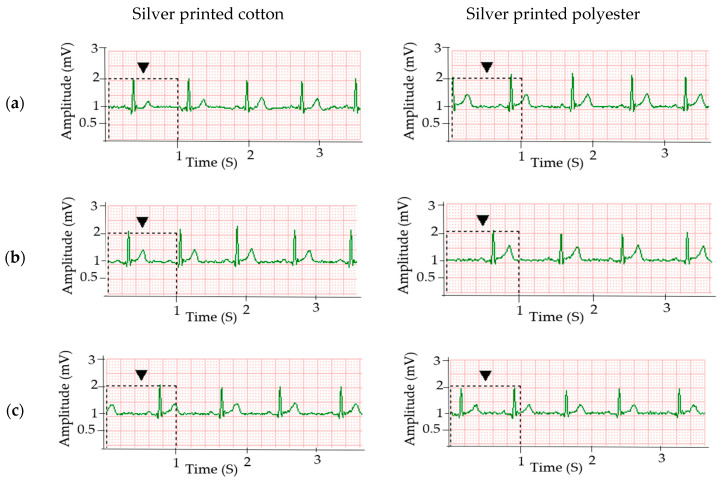
ECG signals collected using silver printed cotton and polyester electrode after (**a**) one washing; (**b**) five washing and (**c**) ten washing cycles.

**Figure 9 sensors-20-06233-f009:**
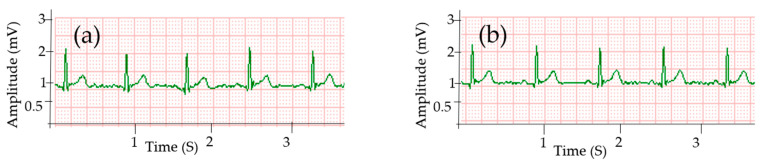
ECG signals collected after six months using; (**a**) silver printed cotton and; (**b**) polyester electrodes.

**Table 1 sensors-20-06233-t001:** Comparison of signals acquired using textile and gelled electrodes, mean and SD over 25 consecutive cycles.

Waveforms and Intervals	Electrode Type		
	Silver Printed Cotton	Silver Printed Polyester	Ag/AgCl
P (mV)	0.09 ± 0.01	0.09 ± 0.02	0.09 ± 0.01
R-peak amplitude (mV)	1.28 ± 0.05	1.26 ± 0.04	1.29 ± 0.05
T (mV)	0.35 ± 0.04	0.35 ± 03	0.35 ± 0.03
PR (ms)	158 ± 4	154 ± 4	158 ± 4
QRS (ms)	93 ± 1	93 ± 1	93 ± 1
QT (ms)	346 ± 8	336 ± 8	348 ± 8
SNR (dB)	33.10 ± 1.31	30.17 ± 1.43	33.52 ± 1.30
Hear rate (bpm)	75 ± 1	73 ± 2	76 ± 1

**Table 2 sensors-20-06233-t002:** HR and R-R interval of textile and gelled electrodes of each subject, mean and SD over 25 consecutive cycles.

Subject	HR (bpm)			R-R Interval (ms)		
	Silver Printed Cotton	Silver Printed Polyester	Ag/AgCl	Silver Printed Cotton	Silver Printed Polyester	Ag/AgCl
1	73	75	76	736 ± 18	756 ± 20	762 ± 16
2	80	79	79	760 ± 23	712 ± 40	736 ± 16
3	82	84	82	761 ± 27	720 ± 27	727 ± 39
4	80	80	82	763 ± 20	768 ± 28	736 ± 20
5	90	92	93	643 ± 12	653 ± 14	641 ± 15
6	78	77	77	797 ± 49	782 ± 53	807 ± 45

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
