# Peer review of "Development of Washable Silver Printed Textile Electrodes for Long-Term ECG Monitoring"

_sensors, 2020, doi:10.3390/s20216233_

Round 1

Reviewer 1 Report

Nice and much needed research. Some moderate remarks:

line 44 - "the ionic current generated by the heart is detected and converted into electrical current", perhaps the ionic is still electrical?!

line 103 - what is GSM?

figure 3, 4, 6, 8 - it would be nice to have axes present with units.

level of skin irritation could be examined as well

abbreviations are defined and redefined rather haphazardly

Reviewer 2 Report

This paper describes the development and test of screen-printed textile electrodes for ECG monitoring. In general it is routine work, but is in scope for the journal and is an active area of research at present.

The authors should comment on:

- The ink is screen-printed through the substrate, i.e. it appears as if both sides of the fabric are coated. Is this correct, and if so, does it have any implications for electrode use or fabrication in a real-world scenario?

-The authors mention that the fabric/electrodes are 'conditioned' before use. Why is this done and how does this process affect performance?

  • an asynchronous approach is taken to measuring ECG on the wrist. How long is elapsed between measurements, and in which order are they performed? Is there a possibility that the skin's stratum corneum barrier is altered or comprimised by the electrodes (e.g. by velcro abrasion, textile abrasion, sweat buildup or most likely gel soakage) ? If so, this is likely to affect signal quality during subsequent measurements unless a period of time is allowed for the skin to recover adequately.
  • - Table 1: consider truncating errors to the same precision as the measurement.

Reviewer 3 Report

The paper deals with the development of washable silver printed textile electrodes for monitoring of ECG. The paper is very well written. The process of the electrode development and electrode testing are clearly described. Although the idea of a silver screen-printed electrode is not new, the paper provides additional information about the developed electrode, which would be interesting for developers dealing with monitoring of ECG using smart textiles. The paper provides an electrical description of electrodes with detailed images from the electron microscope. The testing involves the impact of different skin-electrode pressure and washing cycles on the sensed ECG signal's quality. The results are presented in well-arranged graphs and tables. The results show that the silver printed electrode's ability to sense ECG is similar to standard Ag/AgCl gel electrodes.

Reviewer 4 Report

The work deals with the development and testing of electrodes on textile substrates using screen printing techniques to capture the electrocadiographic signal. Although a review of the state of the art is carried out, the paper does not clearly indicate and show results regarding the advantage and application of the ECG signals picked up with the proposed electrodes with respect to other techniques proposed in the literature or existing on the market.

Other concerns

  • It would be necessary to justify the selection of the silver ink employeed (for instance justify the silver percentatge in the ink selected among others available in the market)
  • Electrode-skin impedance measurements have not been carried out. This is a key measure in Surface bioelectric recording
  • Only one subject has been tested, It would be advisable increase the number of volunteers for testing the elèctrodes.
  • A lower cut-off frequency of 1Hz in the ECG uptake greatly limits the ECG diagnostic capacity, beyond determining the HR? What would be the applicability of the device and its advantage over current solutions on the market?
  • How has the noise been identified to be able to extract the SNR ratio? This must be clarified in Methods section
  • How many electrodes have been tested? Has the normality hypothesis been verified when applying the statistical t-test in the measurements?
  • Has it been verified whether the stretching of the textile substrate causes the surface of the electrode to crack, causing a deterioration in the performance of the electrode, for example by increasing its resistance? Trials should be carried out on this sense
  • The images in figure 3 do not have a temporal or amplitude scale, the same as those in figures 4, 6 and 8. Scales must be included in all plots
  • It is indicated that the utility of the developed electrodes is to pick up long-term ECG records, but no results related to long-term ECG records are attached.
  • What was the level of comfort associated with the different levels of pressure exerted? This is determinant in long-term ECG recordings
  • Numerical values regarding the variation of resistance and skin electrode impedance are required, as well as the ECG recordin characteristics, depending on the number of electrode washes (considering a sufficient number of tested electrodes and subjects)
  • It would be interesting to discuss the performance of the silver screen-printed electrodes on textiles presented in this work with respect to the performance of screen-printed electrodes developed using polymeric inks such as PEDOT: PSS on a textile substrates for ECG recordings in different configurations

Round 2

Reviewer 4 Report

Most concerns and suggestions have been succesfully addressed. I only suggest:

-To remove average values from table 2.

-To change figure 5 taking into account R-peak amplitudes from all subjects

Author Response

This manuscript is a resubmission of an earlier submission. The following is a list of the peer review reports and author responses from that submission.